# Physical Virology in Spain

David Reguera [1], Pedro J. de Pablo [2], Nicola G. A. Abrescia [3], Mauricio G. Mateu [4], Javier Hernández-Rojas [5], José R. Castón [6] and Carmen San Martín [6,*]

1 Departament de Física de la Matèria Condensada and Universitat de Barcelona Institute of Complex Systems (UBICS), Universitat de Barcelona, 08028 Barcelona, Spain; dreguera@ub.edu
2 Departamento de Física de la Materia Condensada and IFIMAC, Universidad Autónoma de Madrid, 28049 Madrid, Spain; p.j.depablo@uam.es
3 Structure and Cell Biology of Viruses Lab, Center for Cooperative Research in Biosciences (CIC bioGUNE), Basque Research and Technology Alliance (BRTA), Bizkaia Technology Park, 48160 Derio, Spain; nabrescia@cicbiogune.es
4 Centro de Biología Molecular "Severo Ochoa" (CSIC-UAM) and Department of Molecular Biology, Universidad Autónoma de Madrid, Campus de Cantoblanco, 28049 Madrid, Spain; mgarcia@cbm.csic.es
5 Instituto Universitario de Estudios Avanzados en Física Atómica, Molecular y Fotónica (IUdEA) and Department of Physics, Universidad de La Laguna, 38200 Santa Cruz de Tenerife, Spain; jhrojas@ull.edu.es
6 Department of Macromolecular Structures, Centro Nacional de Biotecnología (CNB-CSIC), 28049 Madrid, Spain; jrcaston@cnb.csic.es
* Correspondence: carmen@cnb.csic.es

**Abstract:** Virus particles consist of a protein coat that protects their genetic material and delivers it to the host cell for self-replication. Understanding the interplay between virus structure and function is a requirement for understanding critical processes in the infectious cycle such as entry, uncoating, genome metabolism, capsid assembly, maturation, and propagation. Together with well-established techniques in cell and molecular biology, physical virology has emerged as a rapidly developing field, providing detailed, novel information on the basic principles of virus assembly, disassembly, and dynamics. The Spanish research community contains a good number of groups that apply their knowledge on biology, physics, or chemistry to the study of viruses. Some of these groups got together in 2010 under the umbrella of the Spanish Interdisciplinary Network on Virus Biophysics (BioFiViNet). Thirteen years later, the network remains a fertile ground for interdisciplinary collaborations geared to reveal new aspects on the physical properties of virus particles, their role in regulating the infectious cycle, and their exploitation for the development of virus-based nanotechnology tools. Here, we highlight some achievements of Spanish groups in the field of physical virology.

**Keywords:** virus assembly; capsid stability; virus mechanics; nanocontainers; theoretical models; cryo-electron microscopy; atomic force microscopy

## 1. Introduction

Viruses are pathogens causing disease, but they can also be modified to serve as vectors in gene therapy or nanocontainers for technological purposes. Virus capsids are the vehicle for transmission of the viral genome to the next host cell. In simple viruses, the capsid consists of a number of copies of the same or a few different proteins organized into a symmetric oligomer [1]. Structurally complex virus particles present a larger variety of components than simple viruses. They may contain accessory proteins with specific roles or incorporate non-proteic elements such as lipids. Complex capsids present a range of geometrical variability, including symmetry mismatches, singular vertices, or completely asymmetric elements [2]. Virus capsids play multiple roles throughout the infectious cycle, from host recognition to successful genome delivery. To do so, they must be stable enough to survive deleterious environmental conditions, but they must be able to disassemble at the correct time and place to successfully deliver their genome into the host cell. Controlled

uncoating is a critical part of the infectious cycle, as virions must fall apart in an orderly fashion guided by cellular (e.g., receptor binding) or environmental (e.g., pH change) cues [3,4]. To achieve this remarkable stability modulation, viruses must rely on general physical and chemical mechanisms. Understanding how virus particles are organized, how they are assembled, and how they are dismantled once they reach a new host cell is required to understand the basic principles of biological macromolecular assembly and function, facilitate the design of new antiviral drugs or virucidal materials, and engineer vectors for efficient gene delivery.

Even in the absence of their genetic material, virus coat proteins are often able to self-assemble in vitro, and the structural and physical traits of the empty capsids make them ideal candidates for many promising applications [5]. Their tunable and perfectly monodisperse size in the range of tens of nanometers converts viral capsids into invaluable building blocks for patterning on the nanometer scale or scaffolds to template the production of nanowires or electrodes for efficient batteries [6]. Their unique mechanical properties and conformational response to changes in pH or salt concentration offer new ways to develop sensors or molecular switches for nanoelectronics. Moreover, coat proteins can be genetically engineered to functionalize their external or internal surfaces, turning them into customized nanocontainers, nanoreactors [7], or in vivo imaging probes. Efficient encapsulation and subsequent controlled delivery of the cargo at the proper target is also potentially very useful for the selective delivery of drugs or other non-genetic materials. Many of these potential applications depend on the unique properties of the viral capsid self-assembled structure and its ability to encapsulate and release its cargo.

Different techniques such as fluorescence, calorimetry, X-ray crystallography, electron microscopy (EM), or molecular dynamics, have been used to investigate the molecular basis of virus assembly and disassembly. In recent years, cryo-electron microscopy (cryo-EM) has become the technique of choice to study the structure of virus particles at high resolution [8,9] and in the cell [10]. Also, the importance of understanding physical properties of virus capsids has lately been recognized, as a means both to decipher basic macromolecular processes and to exploit them for bio- and nanotechnological applications [11–14]. The application of atomic force microscopy (AFM) has revolutionized the field, providing invaluable information on the physical properties of individual viral particles. Cryo-EM and AFM of viruses and viral capsids are complementary methods with which to examine the close relationship between physical structure and biological function. Whereas cryo-EM facilitates understanding of structure, AFM can provide information not evident from structural data, such as capsid stiffness or topographical evolution in liquid environment [15–17]. These new experimental approaches have raised the need for theoretical and coarse-grained models to fully understand the relationship between mechanical response, stability, and infectivity in viral capsids. Considerable progress has been made in understanding the theoretical aspects of virus assembly and mechanical response, using simulations of various complexities, from coarse grain to full atom [18–20].

Recognizing the importance of interdisciplinary networking, in 2010, several groups in the Spanish scientific community got together under the umbrella of the BioFiViNet ("Red de BioFísica de Virus") project to promote our understanding of the physical aspects of virus assembly, disassembly, and engineering. BioFiViNet has since developed into a vibrant community integrating groups with expertise in theoretical and condensed matter physics, molecular and structural biology, biophysics, and bio- and nanotechnology. A notable contribution of this community was publication of the book *Structure and Physics of Viruses*, which aimed to disseminate this new field among early career researchers [21]. Here, we highlight some of the work developed in recent years by BioFiViNet members.

## 2. Theoretical Modeling of the Assembly and Mechanical Properties of Viral Capsids at Different Levels of Coarse Graining

One of the most fascinating aspects of viruses is the exquisite architecture of their protein shell (the capsid) and the process by which it assembles. Viruses with different

genetic material, different size, and infecting different hosts end up adopting common geometries for their capsids, which in most cases are very well defined in terms of size, shape, geometry, and number of proteins. The catalogue of architectures of viral capsids that are not pleomorphic include: spherical shells with icosahedral symmetry characterized by a discrete triangulation number, the T-number; rod-like viruses with helical symmetry; spherocylindrical or conical capsids; and even the exotic spindle-shaped and bottle-shaped shells discovered in archaeal and extremophile viruses [2,22–24]. The fact that very diverse viruses end up spontaneously adopting a common set of architectures suggests the existence of general and common principles involved in their formation.

The group led by D. Reguera develops theoretical models and simulations at different levels of coarse graining to provide physical insights into the mechanisms underlying the architecture, assembly, and mechanical stability of virus capsids. Simple physical models have been able to explain why spherical viruses adopt icosahedral symmetry [25]. Beyond the clever principles laid down by Caspar and Klug (CK) [26] to explain the geometrical rules for the construction of icosahedral viruses, the physical mechanism justifying the ubiquitous presence of icosahedral symmetry in viral shells was unknown. Using a very low-resolution description of a capsid (capsomers placed on a spherical template) combined with Monte Carlo simulations, it was possible to show that icosahedral symmetry arises naturally from a very general physical principle: the energy minimization of a set of identical units interacting with a potential that includes (i) an attractive contribution to drive the assembly, and (ii) a short-term repulsion to prevent overlapping [25]. If the system is allowed to switch between two capsomer sizes (hexamers and pentamers), the optimal arrangement of N-interacting capsomers occurs for discretized values of N and ends up coinciding with the number and disposition of capsomers in T-number capsids. Moreover, if only one type of structural unit is available, e.g., only pentamers, structures with non-icosahedral symmetry, such as the snub cube adopted by polyomavirus in vitro [27], become energy minima too. Thus, this simple model explains why spherical viruses adopt icosahedral symmetry and the structure of all-pentamer viruses, which were puzzling exceptions to the CK classification.

For non-spherical viruses with architectures resembling a capsule or spherocylinder, the situation was more complicated since there was no complete structural or geometrical classification analogous to that of CK. By assuming that these viruses were hexagonally ordered tubes closed by icosahedral caps cut perpendicular to the 5-fold, 3-fold, and 2-fold axes of symmetry, it was possible to explain the geometrical rules of construction of prolate viruses [28]. Using the same model of interaction between capsomers now placed on top of a spherocylindrical template, it was possible to show that most of these structures, but not all of them, were indeed energetically optimal [29]. For large cylinder diameters, only spherocylindrical viruses with 5-fold axial symmetry were energetically favorable, since fitting a tube to a cap with 3-fold and especially 2-fold symmetries involves very high elastic stresses. With this geometrical and physical model, it was possible to establish a classification of elongated viruses, extending CK's ideas, and the rules for the discretized lengths, diameters, and numbers of proteins allowed for these viruses that open the door to identifying the structures of unknown viruses.

All of these models needed a template to force the capsid to adopt a particular shape. However, assembly can proceed directly from capsid proteins in solution without the presence of any mold or template. To account for that possibility, two more essential components of the interaction between capsid proteins were added to the model: the spontaneous curvature, representing the preferred angle of binding between capsid proteins; and the bending stiffness, quantifying the energy cost needed to bend two proteins out of their preferred angle of interaction. Now the model successfully mimicked the in vitro assembly of empty spherical capsids with different T-numbers [30]. A phase diagram constructed in terms of the two new parameters (Figure 1a) identified the range of stability of the different structures. The diagram shows that the spontaneous curvature is the main parameter selecting the size of the shell, while the bending stiffness mostly dictates whether the resulting

virus is rounded or faceted. More importantly, the model provides important clues about possible mechanisms to destabilize a virus, leading to open, partial, or collapsed structures, which can be of interest as potential therapeutic strategies to stop viral infections.

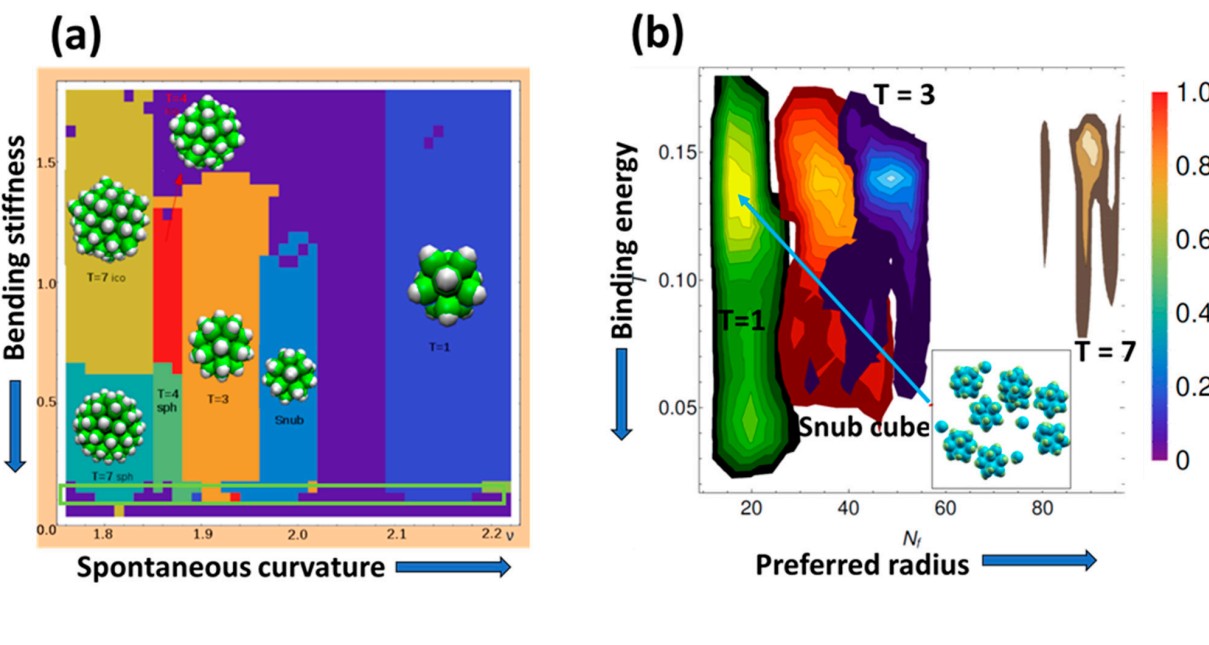

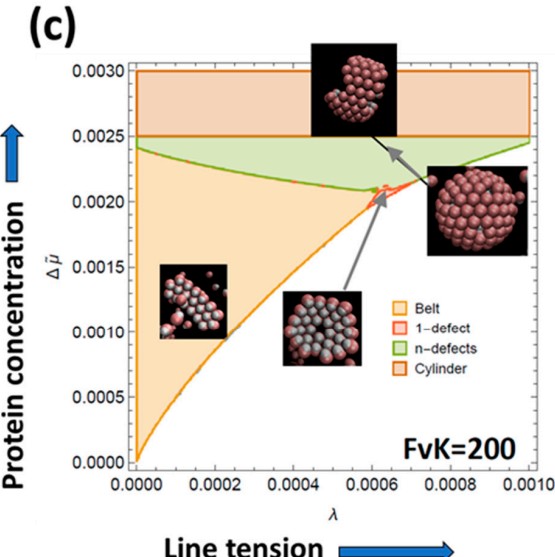

**Figure 1.** Theoretical phase diagrams of viral assembly. (**a**) Representative phase diagram of the region of stability of spherical viral shells made with one type of capsomer as a function of the bending stiffness and spontaneous curvature. The green rectangle highlights the smaller region where assembly is feasible from a kinetic point of view. Adapted with permission from ref. [30]. Copyright 2016 American Chemical Society (**b**) Phase diagram showing the efficiency of the assembly (color scale) of different T-number shells as a function of the binding energy (measured in terms of the reduced temperature T) and preferred radius (measured in terms of the parameter $N_f$ using the model of ref. [31]), Adapted from ref. [32]. (**c**) Phase diagram of the most stable shape of a forming shell as a function of the protein concentration (expressed in terms of a scaled chemical potential) and scaled line tension for a value of the ratio between the elastic and bending energy (i.e., the Flöppl-von Kárman (FvK) number) of 200. Four possible regions are shown, corresponding to the formation of tubes (dark orange), spherical shells (green), ribbons (light orange), and frustrated shells with one defect (red). The insets show the structures observed in simulations. Adapted from ref. [33].

Generalizations of CK geometrical construction have been provided to account for other anomalous icosahedral viral capsids [34,35]. J. Hernández-Rojas and collaborators have recently designed cost functions, the targets of which are the structures observed in known viral capsids [36]. Each target structure is reproduced as the global minimum of the corresponding cost function. With this coarse-grained model, the researchers reproduced practically any capsid structure, including those supported by the models of CK, Twarock and Luque, Rochal and coworkers, double-shell capsids, capsids with protruding decorations, along with many other anomalous structures.

Models have also been developed to analyze the kinetics of virus self-assembly, a non-equilibrium process similar to a phase transition [37]. Brownian dynamics simulations showed that the range of conditions required for successful assembly are different from the ones that provide optimal energy or at which an already formed capsid is stable [32]. Efficient assembly occurs in a very narrow range of conditions (Figure 1a) and requires weak energies (on the order of 5–7 $k_B$T) and spontaneous curvatures larger than the optimal value (Figure 1b) to prevent premature closure while the capsid is growing [38]. It was also found that the outcome of viral assembly can be recast into a universal phase diagram that explains how competition between bending and elastic contributions dictates the shape of viral capsids (Figure 1c) [33]. By modifying the assembly conditions, a spherical virus can become trapped into a partial shell or ribbon-like structure or end up forming tubes instead of spheres, which will compromise its viability. This competition also explains why some viruses need a maturation step, since capsids with high mechanical resistance cannot be self-assembled directly as spherical structures. The results of these studies suggest promising routes to hinder viral infections by inducing misassembly.

Other simple models have been designed to analyze the normal mode frequencies of icosahedral empty viral capsids. Namely, ref. [39] proposed a coarse-grained model for the interaction energy of viral capsids to evaluate the normal-mode spectrum of an icosahedral T = 1 capsid. The model performance was evaluated by fitting its predicted spectrum to the full-atom results for the Satellite Tobacco Necrosis Virus (STNV) capsid [40]. This model found a latent instability consistent with the current knowledge about the STNV capsid, which does not self-assemble in the absence of RNA.

The same simple models that have been used to describe the shape and assembly of viruses are very useful for understanding different aspects of their mechanical response (see Section 3). Simple continuum elasticity can be used to infer viral mechanical properties (e.g., Young's modulus) [41] and the pressure generated by the genetic material [42–45] or other cargos encapsulated inside [41]. Describing viruses with a discretized model in terms of capsomers explains the anisotropic response [46,47], the breaking patterns of viruses [43], or the existence of pre-stress [48]. These models are, in general, extremely useful for understanding the behavior of viruses in AFM experiments [49]. Even simple models of dsDNA as an elastic rod provide insightful information about how the genetic material of a virus is packed inside the capsid or released efficiently into bacterial host cells [50].

In summary, theoretical models and coarse-grained simulations are very useful for understanding different aspects of viruses, such as how they self-assemble, their mechanical properties, or the way they package and deliver their genetic material. This understanding opens the door to developing alternative strategies to fight viral infections based on interfering with these basic processes. There are still many biophysical aspects to discover by modeling the different steps of the replication cycle of viruses, which can help us to better understand how they work and how to be ready for fighting future pandemics.

## 3. Physical Virology with Atomic Force Microscopy

The study of viruses usually involves bulk experiments, which gather average data from a large number of particles, thus considering them indistinguishable. However, biochemical processes are highly asynchronous and intermediate states are poorly populated. Therefore, average measurements might conceal details of the processes taking place in

individual entities composing the bulk specimen. Atomic Force Microscopy (AFM) enables imaging and manipulating individual virus particles adsorbed on a surface in liquid milieu by using a sharp tip (~10 nm) located at the end of a microcantilever (Figure 2a) [15].

In AFM, the tip mounted at the end of a flexible microcantilever scans the sample in X, Y, and Z directions using piezo actuators. While X and Y scanners move the tip over a square region, the interaction with the surface topography will lead to a bending of the cantilever. A four-quadrant photodiode registers the reflection of a laser beam focused at the end of the cantilever, monitoring its deflection. Thus, each pixel of the image located at a position of planar coordinates (X, Y) will be associated with certain bending value of the cantilever. As the tip scans the surface, dragging forces are created that could damage the specimen under study. Methods developed to avoid this damage include the so-called jumping mode (JM) [51]. In JM, the tip moves laterally when it is not in mechanical contact with the sample, thereby almost completely avoiding dragging forces. JM performs consecutive approach–release cycles at every pixel of the sample. In each cycle, known as a force vs. distance (FZ) curve, the Z-piezo moves the tip to the sample until establishing mechanical contact and reaching a certain feedback force (Figure 2b, top). After a few milliseconds, the Z-piezo retracts the tip for about 100 nm, releasing the surface. Subsequently, the scanner moves laterally to the next pixel, and the process repeats [51,52]. Especially in liquid milieu, where adhesion forces between the tip and sample are practically absent, JM has proven very successful [15].

The study of virus mechanics by AFM has largely relied on single indentation assays (Figure 2b). In this technique, the Z-piezo moves the tip towards the upper surface of the virus particle until establishing mechanical contact and reaching a certain feedback force, recording an FZ curve (Figure 2b, α). The cantilever bending can be converted into a force, while the indentation of the virus particle is given by the motion of the Z-piezo minus the bending of the cantilever. After contact between the tip and particle is established, the FZ curves normally show an approximate linear behavior, which corresponds to the elastic regime of the protein cage, from which the particle elastic constant $k$ can be derived (Figure 2b, β). When the indentation depth is kept small, it is possible to record repetitive FZ curves, and the particle elastically deforms in a reversible way. However, when the indentation surpasses the elastics limit, the particle breaks (Figure 2b, γ). In this case, the curve provides information on both the breaking force and critical indentation tolerated by the capsid, and the image recorded afterwards provides information about the capsid's weakest points (Figure 2b, bottom) and breaking pattern.

The breaking force describes the maximal force that a protein cage can sustain before it collapses, exhibiting large structural alterations. Therefore, single nanoindentation assays at forces beyond the elastic limit are not appropriate to investigate the sequential capsid disassembly processes that may be relevant for the virus infectious cycle. JM imaging performs a load cycle at each pixel of the virus particle during imaging, reaching about 100 pN [52] and transferring ~10 $k_B$T to the particle at every cycle [43]. This energy is about 10 times the value provided by the impact of a macromolecule, according to the equipartition theorem [53]. Consequently, JM imaging probes the resistance of the virus particle to mechanical inputs of moderate energies. Repeated imaging of the same virus capsid under these conditions results in a gradual, increasing accumulation of mechanical stress, which allows sequential disassembly of virus capsids under mechanical fatigue to be simultaneously induced and monitored. Such mechanical fatigue experiments have reproduced the natural pathway of adenovirus uncoating much more accurately than single nano-indentation experiments at higher forces (see Section 6) [54].

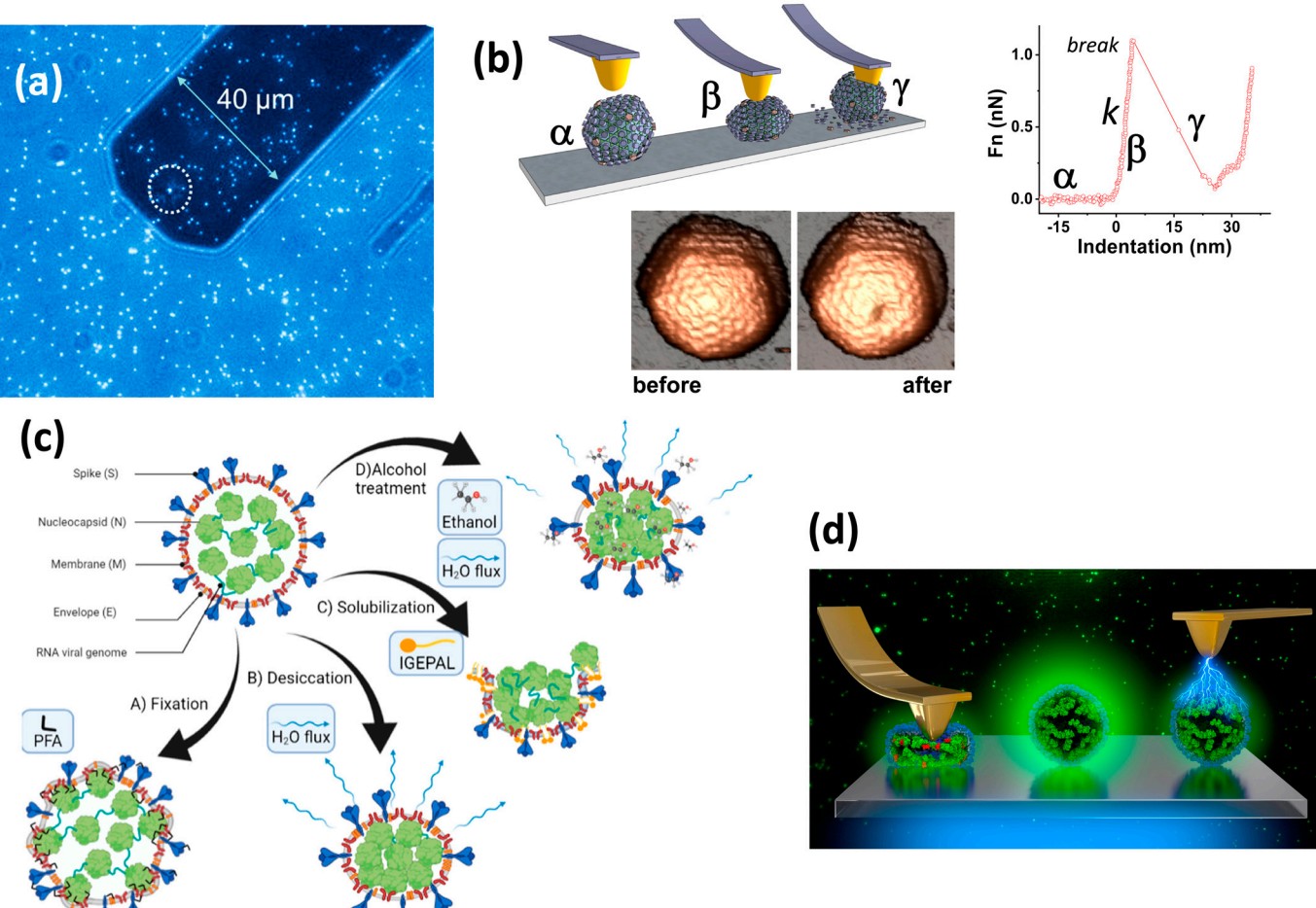

**Figure 2.** Atomic force microscopy of viruses. (**a**) Geometrical comparison between the cantilever with the tip at the end (white circle) and virus capsids of ~100 nm in size (white blobs). Reproduced from ref. [55] with permission. (**b**) (**Top left**) The three stages of a virus particle nanoindentation experiment. α: approach, β: indentation, γ: breakage. (**Top right**) A FZ chart with the same keys (α, β, γ) as the cartoon at top left. (**Bottom**) the FZ effects on the P22 virus capsid. Bottom panel adapted from ref. [56] with permission. (**c**) Cartoon showing the structural progression of TGEV particles under stringent environmental conditions. Reproduced from ref. [57]. (**d**) Artistic rendition for the quenching of GFP packed inside P22 phage induced by insulating (**left**) or conductive (**right**) AFM tips.

The group led by P.J. de Pablo has pioneered the use of AFM nanoindentation and fatigue assays to reveal the remarkable mechanical properties of many different virus particles [15]. These include a Young's Modulus of 3 GPa (close to that of Plexiglas$^{TM}$) and critical strains of ~20% [41]. Thus, a protein virus capsid can elastically deform to about 0.2 times its size (tens of nanometers) before fracturing (Figure 2b, bottom). These properties are intimately related to capsid resistance in stringent environments. For instance, in tailed phages, the structural transformations undergone by virus particles during maturation required for infectivity are accompanied by an increase in the mechanical resistance of the capsid [43]. Maturation can also imply the binding of cementing proteins to the capsid shell, improving both the mechanical and chemical resistance of the cage [53]. Virus mechanics also depend on the cargo stored inside and its interaction with the protein shell, as shown for human adenovirus, where maturation decreases capsid stability rather than increasing it (see Section 6) [45,54,58,59].

Coronaviruses have been lurking pathogens causing occasional outbreaks (SARS-CoV, MERS-CoV) until finally becoming the cause of the recent COVID-19 pandemic [60].

The demanding biosafety requirements for the study of these serious human respiratory pathogens require thorough adaptations of the AFM workflow [61]. An alternative is to use surrogate models to facilitate their investigation. Thus, transmissible gastroenteritis virus (TGEV), a porcine coronavirus, was used to study coronavirus mechanical properties and nanostructure in liquid milieu and its response to different chemical agents commonly used as biocides [57]. Monitoring the structure of single TGEV particles in the presence of detergent and alcohol in real time revealed the stages of gradual degradation of the virus structure in real time. Further, this work indicated that detergent is three orders of magnitude more efficient than alcohol in destabilizing TGEV particles, paving the way for optimizing hygienic protocols for viruses with similar structure, such as SARS-CoV-2 (Figure 2c).

Packing biomolecules inside virus capsids has opened new avenues for the study of molecular function in confined environments. These systems not only mimic the highly crowded conditions in nature but also allow their manipulation at the nanoscale for technological applications. Virus-like particles derived from P22 bacteriophage procapsids packing green fluorescent protein (GFP) have been explored to monitor their emission signals using total internal reflection fluorescence (TIRF) microscopy while simultaneously changing the microenvironment with the stylus of AFM [62]. The mechanical and electronic quenching can be decoupled using insulator and conductive tips, respectively (Figure 2d). Upon indentation with conductive tips, the fluorescence quenches and then recovers regardless of the structural integrity of the capsid. However, with insulator tips, quenching only occurs if GFP molecules remain organized inside the capsid. Electronic quenching is associated with coupling of the protein fluorescence emission with the tip's surface plasmon resonance. In turn, mechanical quenching is a consequence of the unfolding of the aggregated proteins during mechanical disruption of the capsid.

## 4. Assembly, Stability, Dynamics, and Mechanics of Simple Viruses

The pathways by which structurally simple virus capsids are assembled from their building blocks (CBB) had been difficult to unveil experimentally. The group led by M.G. Mateu has studied the self-assembly and disassembly pathways of the icosahedral T = 1 min virus of mice (MVM) capsid in vitro, using a combination of EM and AFM [63,64]. The results revealed that trimeric CBBs are sequentially bound to the growing viral particle in a fully reversible process, and that pentamers of CBBs and capsids lacking one CBB constitute conspicuous assembly intermediates [63].

The thermodynamics and kinetics of assembly and disassembly of the hexagonal capsid protein (CP) lattice that forms the mature human immunodeficiency virus (HIV-1) capsid were also analyzed in vitro by AFM [65,66]. This simplified system and the use of high-speed AFM (HS-AFM) allowed visualization in real time and with single-molecule resolution the pathways by which a virus capsid lattice is assembled (Figure 3) [67]. Individual molecules of the mature HIV-1 capsid protein CA were observed while they built the capsid lattice. Lattice patches grew independently from separate nucleation events through addition of CA molecules or small CA assemblies through many different pathways. The experimental results of the studies with MVM and HIV-1 capsids summarized above verified fundamental aspects of virus capsid assembly predicted by theory or molecular dynamics simulations. Additionally, studies by this group have also revealed the importance of macromolecular crowding for assembly of a virus capsid (the mature HIV-1 capsid) under physiological conditions [68,69], and its inhibition by small molecules [70].

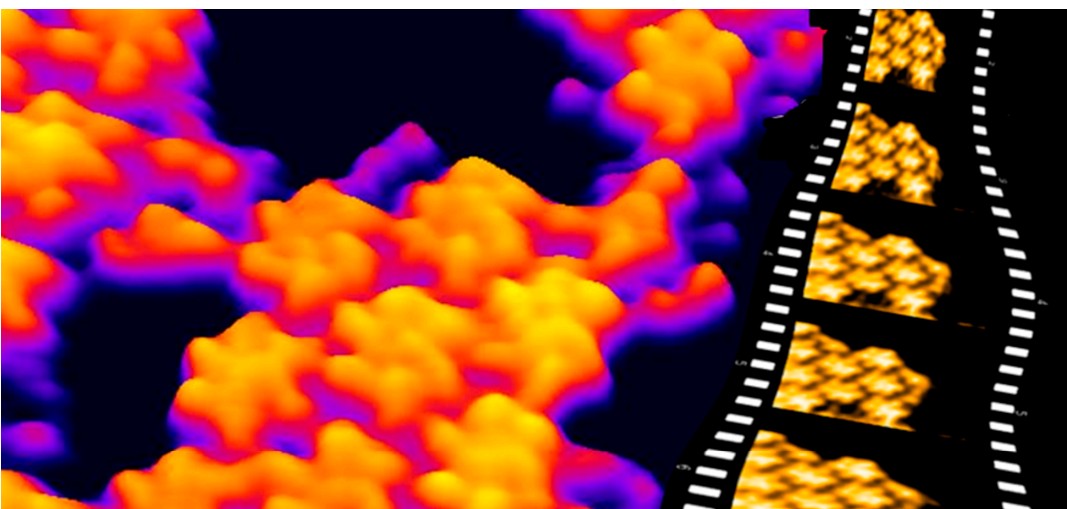

**Figure 3.** Self-assembly of the mature HIV-1 capsid protein lattice on a negatively charged substrate. (**Left**): HS-AFM image of a growing lattice patch. Several capsid protein hexamers are fully assembled, whereas other hexamers are still being formed. (**Right**): Several consecutive HS-AFM images extracted from a movie where assembly of the HIV-1 capsid protein lattice from individual capsid protein molecules is visualized at high resolution in real time. Figure composed by Alejandro Valbuena, CBMSO.

Numerous mutational studies carried out by M. G. Mateu and collaborators have identified individual capsid amino acid residues that play important roles in virus particle assembly, physical stability, conformational transitions, or equilibrium dynamics. These studies provide detailed insight into the complex relationships between virus structure, stability, dynamics, and biological function [4,71]. This group is also using AFM for studying in atomic detail the structural determinants of the mechanical behaviors of viruses, including stiffness, brittleness, strength against disruption by mechanical force, and resistance to material fatigue; the relationships between virus mechanics and virus stability or dynamics; and the biological relevance of virus mechanics [72,73].

The individual effects on MVM mechanics of close to 50 engineered amino acid substitutions in different capsid regions have been analyzed so far. The results show that very small structural changes [74,75] may be translated into remarkable changes in the mechanical strength of the particle [76]. Small structural changes can also lead to large variations in stiffness and intrinsic elasticity in small spherical viruses [75,77–80]. In several cases, changes in the mechanical behaviors of mutant particles were linked to different mechanisms impairing infectivity.

Extensive mutational studies of independent, infectivity-related conformational transitions in MVM unveiled an intimate correlation between the stiffening of capsid regions involved in the transition, and impairment of the transition itself [77,78]. An exponential relationship between the transition rate constant and elastic constant was observed. This relationship was justified in light of transition state theory [78]. In addition, mutational and ligand-binding studies using the MVM capsid [74,75], human rhinovirus capsid [80], or HIV-1 capsid lattice [81,82] unveiled in every case a correlation between changes in intrinsic elasticity and changes in equilibrium dynamics (breathing). Local or global changes in breathing (determined by, for example, HDX-MS) [83] or stiffness (determined by AFM) are different manifestations of the same physical phenomenon: a structure-based variation in the energy required to move atomic groups and larger structural elements in a virus particle a certain distance around their equilibrium positions.

The mechanical properties of virus particles play a role in the biology of the virus: the observed anisotropic distribution of stiffness in the MVM virion [84] may be the result of biological adaptation. Capsid-bound ssDNA segments close to the 2-fold and 3-fold

symmetry axes, but not to the 5-fold region, act like molecular buttresses that mechanically stiffen the capsid or impair its conformational dynamics [78,85]. A heat/force-induced transition that may lead to untimely DNA release is thus impaired, increasing the chances of virion survival. Vertex regions around the capsid pores, free from bound DNA, preserve low stiffness and high conformational dynamics. This feature facilitates an infectivity-determining transition that, in vivo, may be mechanically induced by the encapsidated ssDNA molecule and is required for externalization of infectivity-determining peptide signals through the capsid pores [77].

From an applied perspective, the studies by Mateu and collaborators on virus assembly, stability, dynamics, and mechanics are contributing to the development of: (i) improved virus-inactivated vaccines for foot-and-mouth disease virus based on engineered virions of increased thermostability [86,87]; (ii) peptide-based inhibitors of HIV-1 capsid assembly as lead compounds for the development of novel anti-HIV-1 drugs [88]; antiviral agents based on the modification of virion stiffness or mechanical strength [80,82]; as well as engineered bidimensional virus-based protein materials for biomedical or nanotechnological uses that feature genetically improved thermostability, resistance to mechanical stress and fatigue, and self-healing potential, without unwanted stiffening [81].

## 5. Structural Studies for the Characterization of Virus-Based Nanocontainers

Structural, physical, and functional studies across virus families have provided data that identify the principles of their structure–function relationship. Viral capsid structures and properties have shown the dynamic nature of these macromolecular assemblies along their infectious cycles. In virus-based biotechnology, 3D cryo-EM analysis of viral capsids and virus-like particles (VLPs) is a key tool for the development of improved viral assemblies [89], and structural "virotechnology" (or virus-based technology) is becoming a reality [90].

Hollow protein containers are widespread in nature and include not only viral capsids but also nanocages of bacterial or eukaryotic origin [91–94]. Their inner and outer surfaces can be modified chemically or genetically, and the internal cavity can be used to template, store, and/or arrange molecular cargos [5]. In viruses, the modified capsid proteins can often self-assemble in vitro into non-infectious VLPs. Viral assembly is usually promoted by single-stranded (ss) nucleic acids or by scaffolding proteins (SPs) or membranes (or a combination of these). These natural assembly mechanisms can be used to encapsulate heterologous cargos, including chemicals, enzymes, and/or nucleic acids for a variety of nanotechnological applications [8], such as customized nanoreactors [95], delivery or display vehicles [96–98], devices for imaging [99], and artificial compartments [100] for a variety of applications in nanomedicine, materials science, and synthetic biology.

Here, we highlight two model viral systems used extensively for these applications: the plant virus cowpea chlorotic mottle virus (CCMV), a ssRNA virus that co-assembles with the viral genome, and bacteriophage P22, a dsDNA bacteriophage that requires SP [13]. In both cases, cryo-EM analysis carried out by J.R. Castón's group was important for understanding structural and functional differences in hybrid assemblies. CCMV has a 280 Å diameter T = 3 capsid enclosing a 180 Å diameter cavity, and it can be produced in gram quantities in infected plants. P22 VLPs assemble into a T = 7 procapsid (600 Å outer diameter) that expands to a mature capsid (650 Å diameter) after heating in vitro.

Whereas the strategy for P22 is based on fusion of a heterologous protein to the SP C-terminal helix-loop-helix fragment that binds with high affinity to the capsid interior [41,101], for CCMV, selected protein/enzymes are bound covalently to nucleic acids [102,103] (Figure 4). Both the scaffolding fragment and nucleic acid segment act as efficient bait for specific regions on the CP inner surface. In CCMV, heterologous anionic cargos, which also emulate the viral genome, act as a template for CP assembly into different icosahedral (with T = 1, 2, or 3) and tubular structures [103].

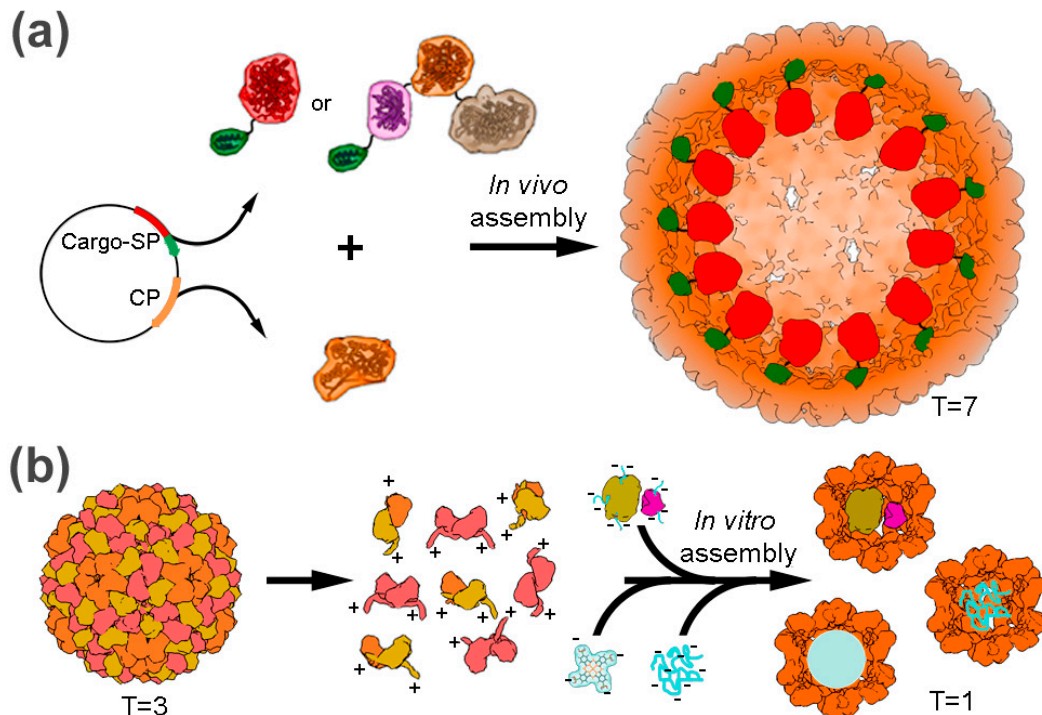

**Figure 4.** Assembly strategies to develop hybrid P22 and CCMV particles. (**a**) In vivo assembly of P22 VLP with different cargos. Coexpression in *Escherichia coli* of capsid protein (CP, orange) with N-terminal-truncated SP (green) fused to different cargos results in assembly of 60 nm T = 7 procapsids with encapsulated cargo. The SP fusion strategy has been used to incorporate from one to three fused enzymes/proteins. (**b**) In vitro assembly of CCMV. The CCMV T = 3 capsid is disassembled into CP dimers. After removal of viral RNA, CP re-assembly requires negative cargos as templates (instead of the viral ssRNA) and a variety of structures can be formed, such as tubes or icosahedral capsids of different sizes (only 18 nm T = 1 capsids are shown). Encapsulation is driven by electrostatic interactions between the polyanionic cargo (blue) and positively charged residues from the CP N-terminal region (reproduced with permission from ref. [8]).

Bacterial nanocages, termed encapsulins, are nanoplatforms with many similarities to the P22-like system [104]. The *Brevibacterium linens* encapsulin houses within its cavity a dye-decolorizing peroxidase (DyP) involved in oxidative stress. Like the SP of P22, the DyP encapsulation mechanism is mediated by its C-terminal end, which interacts specifically with a defined inner surface of the encapsulin region (similar to a capsid protein). Fusion of the DyP C-terminal end to a heterologous protein (teal fluorescent protein) allows it to be packaged in these bacterial nanocontainers [105].

New approaches in cryo-EM extend these analyses to the asymmetric organization of the packaged genome and minor structural proteins, such as viral polymerases, that were previously missed. Asymmetric structures have important functions in many steps of the virus replication cycle, and many of these will be key targets for the development of new antiviral drugs [106–108].

## 6. Assembly and Disassembly of Complex Viruses: The Case of Adenovirus

As the complexity of virus particles increases, the combination of structural and physical approaches becomes even more necessary to understand the morphogenesis and dynamics of viruses where, too often, no in vitro assembly system is available. An example of such a specimen is the adenovirus (AdV) particle. AdVs are relevant as both pathogens and therapeutic vectors in human and animal health [109,110]. The family prototype, human AdV type 5 (Ad5), has a 35 kbp dsDNA genome housed inside a 95 nm *pseudo* T = 25 icosahedral capsid. Human AdV with low seroprevalence, as well as non-

human AdVs, have been proposed as vector candidates to circumvent targeting problems posed by Ad5. Thanks to advances in cryo-EM that have greatly facilitated structural determination of these large biological macromolecules, the architectures of some of these vector candidates have now been examined [111–117].

The AdV icosahedral facets are formed by trimers of the major coat protein, hexon. A pentamer of penton base protein sits at each vertex, in complex with a trimer of the receptor-binding fiber. In addition, in Ad5, correct assembly requires four different minor coat proteins: IIIa, VI, and VIII on the inner capsid surface, and IX outside (reviewed in [118,119]) (Figure 5a). In the non-icosahedral core, the genome is packed together with ~25 MDa of positively charged, virus-encoded proteins V, VII, and μ. The AdV protease (AVP) is also packaged together with the genome. Except for AVP, there are practically no structural data on any of the core proteins. Based on a combination of cryo-electron tomography (cryo-ET) and molecular dynamics, the group led by C. San Martín proposed a model where the core would be organized by interactions similar to those in a fluid of soft repulsive particles [120].

Ad5 virions enter the cell via endocytosis to the early endosome. During attachment and internalization, the virion undergoes stepwise disassembly that starts by releasing penton proteins, followed by the internal minor coat protein VI, a membrane-lytic factor required for endosome escape. Cleavage of several capsid and core proteins by AVP is required to yield the mature, infectious virion [121,122]. Immature Ad5 particles are defective in the initial stages of uncoating. They do not expose the lytic protein VI in the early endosome, becoming trapped in the endocytic pathway, and infection is aborted. After endosome escape, partially disrupted Ad5 particles are transported along the microtubules to the nuclear pore, where dsDNA enters the nucleus via unknown mechanisms that might be related to the condensation state of dsDNA [123]. Collaborative studies by the groups of P.J. de Pablo and C. San Martín have shown how modifications in core proteins change virion structure, mechanics, and stability, and, therefore, uncoating and infectivity.

In vitro capsid disruption assays, cryo-EM, and AFM analyses showed that maturation cleavages disrupt interactions in both the capsid and core that stabilize the immature particle. Maturation makes Ad5 metastable, priming the virion for uncoating by facilitating vertex release and loosening the condensed genome and its attachment to the icosahedral shell [58,59]. Mechanical fatigue induces stepwise disassembly of mature and immature AdV particles, mimicking their uncoating differences in the cell [45,52,54,124,125] (Figure 5b). Maturation of core proteins VII and μ increases the internal pressure exerted by the genome, providing a physical mechanism to facilitate penton release (Figure 5c) [45]. Cryo-ET analyses of heated or acidified Ad5 particles, as well as simultaneous AFM/TIRF imaging after cracking single Ad5 particles with the AFM (mechanical unpacking), showed that the genome is less condensed after maturation and diffuses more easily out of the open capsids [58,59,124]. One of the five Ad5 proteins required for genome packaging, L1 52/55k, is also a substrate for AVP [126]. L1 52/55k acts as a "velcro" tether, recruiting capsid fragments to the condensed genome during assembly [127]. Proteolytic cleavages disrupt this tether, facilitating separation of the genome from capsid fragments in the final stages of uncoating [126,128]. Recent studies have shown that L1 52/55k has a liquid-liquid phase separation activity that is required for Ad5 morphogenesis [129].

Cryo-EM and AFM combined are also helping to elucidate the specific role of each one of the core proteins in the Ad5 infectious cycle. Core protein V stabilizes the capsid to avoid genome misdelivery in the cytosol and facilitates genome release upon particle disassembly at the nuclear pore [130,131]. Protein VII modulates capsid internal pressure and DNA release from partially disrupted particles, which could be crucial for keeping the genome protected while the particle travels to the nuclear pore [44]. Competition between proteins VII and VI for hexon binding during assembly is required to facilitate the complete maturation of VI and its exposure during uncoating in the endosome [132].

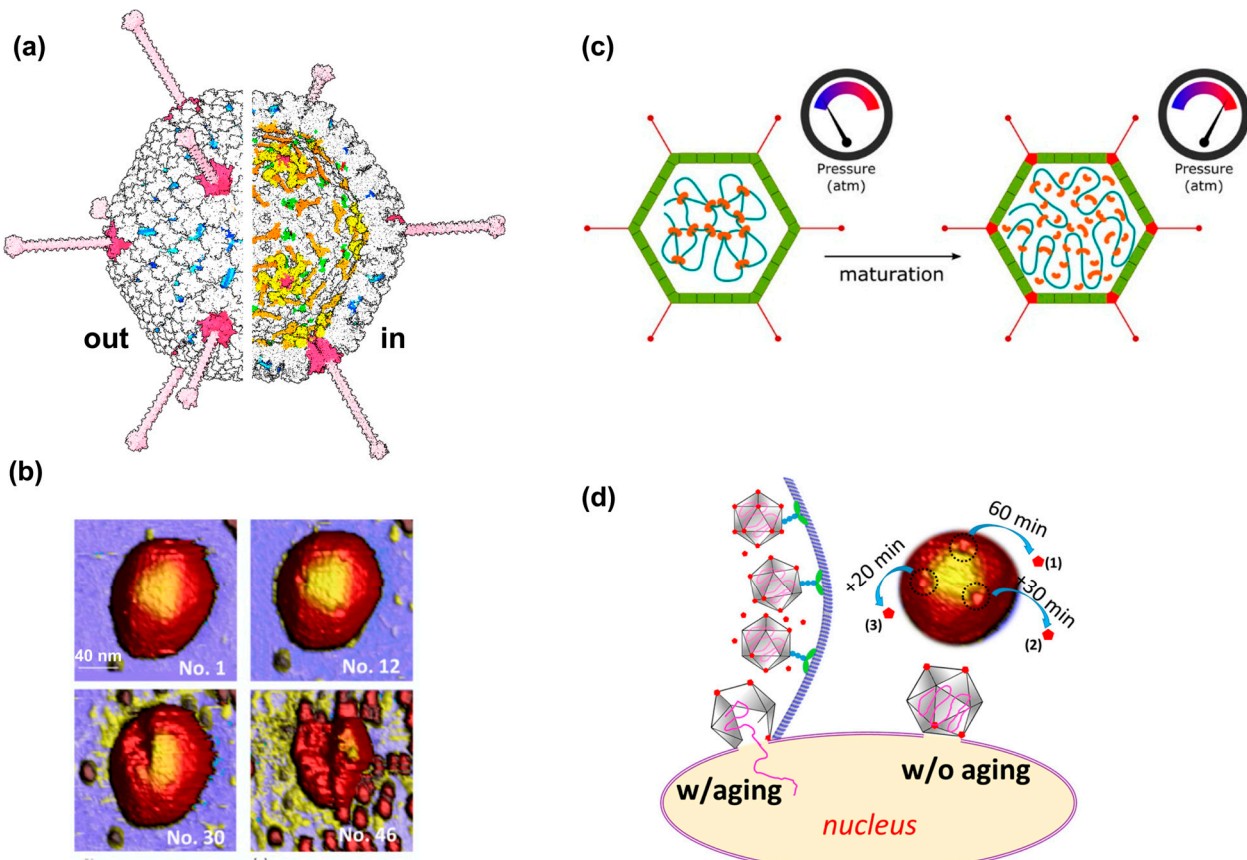

**Figure 5.** Insights into adenovirus assembly and disassembly from physical virology approaches. (**a**) General view of the Ad5 architecture as reported in ref. [116]. The fibers, which are not resolved in icosahedrally averaged maps, are modeled from the crystal structure of the knob and distal shaft [133]. The right half of the figure is a cut open view showing the internal minor capsid proteins. Hexon is depicted in white; penton base in dark pink; fibers in light pink; protein IX in blue and cyan; IIIa in yellow; VIII in orange, and VI in green. (**b**) AFM images acquired during a mechanical fatigue assay showing the sequential disassembly of an Ad5 particle. (**Top-left**): Intact particle. (**Top-right**): The same particle with three pentons lost. (**Bottom-left**): The same particle after the shell has cracked and its contents start being released (yellow filaments on the blue surface). (**Bottom-right**): Final state of the collapsed particle. The number of consecutive images taken before reaching each disassembly step is indicated in each panel. Reproduced from ref. [125]. (**c**) Uncleaved core proteins in the immature Ad5 particle act as DNA condensing agents. Upon AVP cleavage, pressurization induced by decompaction of the viral genome destabilizes pentons, facilitating initiation of stepwise disassembly and enabling escape from the endosome. Reprinted with permission from ref. [45]. Copyright 2015 American Chemical Society. (**d**) Penton aging revealed by real-time monitoring of Ad5 disruption by mechanical fatigue at different forces. Aging guarantees that the particle is ready to deliver the genome at the nuclear pore, after initial disruption during endocytosis and transport along microtubules. Modified from ref. [125].

Real-time imaging of Ad5 sequential disassembly induced by mechanical fatigue at different forces revealed the dynamics of penton release [125]. Viruses retaining too many pentons would not be able to release their DNA upon docking at the nuclear pore, whereas those with too few pentons would suffer premature genome loss in the cytosol. Mechanical fatigue induced by AFM imaging mimics the stresses sustained by the virus particle during its journey to the nucleus due to molecular crowding and kinesin power strokes. Survival analysis showed that the loss of one penton increased the release rate of the remaining ones, accelerating the overall penton escape rate by ~50% with respect to a

sequence of independent escape events. This "aging" effect, demonstrated for the first time in a biomolecular assembly, provides evidence of a cooperative process involving pentons as far apart as 45 nm and prepares the partially disrupted particles for successful DNA translocation through the nuclear pore (Figure 5d).

## 7. Contributions to the Study of Membrane-Containing Bacteriophage PRD1

Some virus particles possess an internal membrane vesicle, such as the lipid-containing bacteriophage PRD1, which over the years has become the model system for this type of virus. Despite the advances in understanding PRD1 assembly provided by a combination of protein crystallography and virus cryo-EM [134,135], only the X-ray structure at 4.2 Å resolution of the complete particle unveiled the network of protein interactions and mechanism of particle formation [136,137]. Until today, PRD1 is the only membrane-containing virus to have been elucidated by X-ray crystallography. The crystal structure showed how the 720 copies of the major capsid protein P3 arrange and interact on the membrane vesicle, as well as the membrane organization and its interactions with viral proteins and dsDNA. Four capsomers, each formed by one P3 trimer, plus one copy of the penton protein (P31), make up the asymmetric unit in the *pseudo* T = 25 capsid, using the same arrangement as that of the adenovirus capsid. The high-resolution structure of PRD1 provided insight into its assembly by offering a distinctive and scalable mechanism to determine particle size using a molecular tape measure. The tape measure protein, P30, is an elongated molecule with little secondary structure that spans the icosahedron edge from one vertex to the nearest 2-fold axis, where it intertwines with another P30 molecule to define the facet dimensions. The similarities in capsomer fold and organization between PRD1 and human adenovirus prompted the possibility that viruses infecting very different hosts could be classified into so-called structural lineages [138]. Indeed, it is recognized today that numerous viruses across the biosphere belong to what is now called the *Bamfordvirae* kingdom [139].

Since the tail-less bacteriophage PRD1 has an additional structural component in the form of the protein-rich membrane vesicle, considerable effort has been made to understand how dsDNA is packed and how it is translocated into the host cell during infection. Early negative-staining 2D images of PRD1 showed a tubular appendage protruding from one of the twelve icosahedral vertices [140]. The icosahedral ordered protein components of PRD1 were investigated in detail using averaging approaches, but all other components that did not adhere to icosahedral symmetry remained hidden. Only when asymmetric reconstruction of a single particle was performed via cryo-EM could the position and structure of the unique vertex be elucidated [141]. Using immuno-labeling, cryo-EM and cryo-ET techniques targeting mature PRD1, PRD1 procapsid (defective in genome-packaging ATPase P9 protein), and virus-infected cells, the Abrescia Lab reported a new principle of virus–cell interaction necessary for viral genome translocation [142]. According to this study, the tube used for DNA ejection in PRD1 is structured and protrudes from the same vertex that is utilized for DNA packaging. These findings support the direct involvement of membrane-associated proteins that self-assemble and form lattices in the conformational changes required to form the tube. Likely candidates are the proteins P7, P14, P18, and P32 [143]. Notably, the packaged genome does not play a role in the initiation or formation of the tail tube. However, in the procapsid in which the membrane vesicle is devoid of genome, the formation of the tail tube fails to orient correctly and, as a result, the tubular structure does not always protrude from the apex. Furthermore, the genome-free membrane vesicle showed different morphologies from stomatocytes to discocytes, displaying not only the elasto-mechanical properties of unilamellar giant proteo-vesicles but also the triggering factors leading to membrane remodeling [142]. These factors have been attributed to changes in the structural, environmental, and/or osmotic conditions of the inner membrane vesicle caused by phage-binding to the cell receptor (which is still unknown). This event (or series of events) would initially modify the structure of the vertex complex, promoting vertex decapping and exposing the membrane to extracellular milieu

conditions. This would then initiate membrane vesicle remodeling and tube formation. In situ cryo-ET captured PRD1 infecting an *E. coli* cell by puncturing the bacterial cell membrane with the tail tube, elucidating the mechanisms that govern tube self-assembly, viral DNA ejection apparatus, and cell infection [142] (Figure 6). In light of these results, a further study using modeling techniques proposed that the elastic properties of the membrane may favor translocation of the viral genome into the host cell in the early phases of infection [50]. This hypothesis led also to question whether the PRD1 membrane vesicle provides further protection to the viral genome. To protect the genetic cargo outside the cell, viruses have developed various viral protein folds, architectures, and material compositions (protein, lipid, or a mixture of both). This mix of materials and architectures provides specific mechanical properties that meet protection requirements. Indeed, when virions leave the host cell, they may experience extreme environmental conditions.

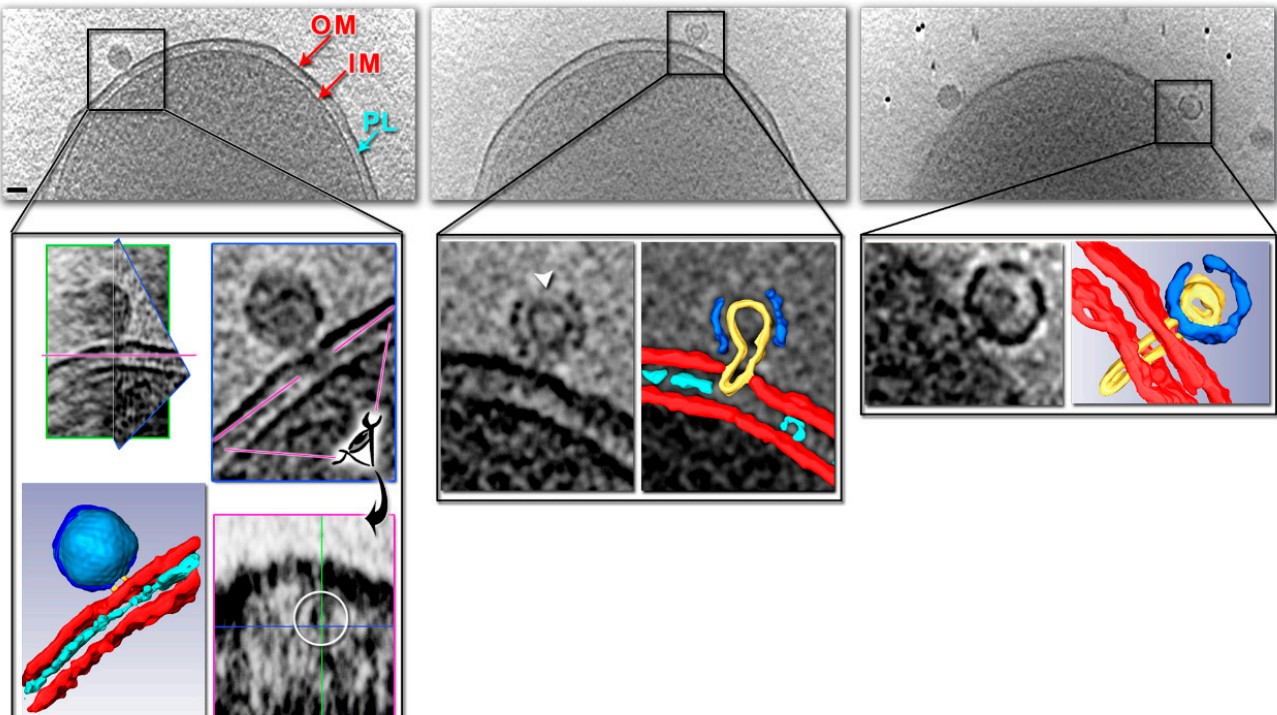

**Figure 6.** In vivo PRD1 genome delivery. Tomographic slices of an *Escherichia coli* cell infected by wild-type PRD1 (∼30 min post infection). Scale bar, 50 nm. (**Left**) Full particle, marked with a black square, with forming tube crossing the outer membrane (OM). Inner plasma membrane (IM) and peptidoglycan layer (PL) are indicated. (**Centre**) Semi-full particle, marked as previous. (**Right**) An almost empty particle, outlined in a black square. Insets provide enlarged views and corresponding segmentations color-coded as follows: blue, capsid; yellow, membrane vesicle and tube; red, outer and inner membranes; cyan, peptidoglycan layer. The white arrowhead indicates additional openings of the capsid. Adapted from ref. [142].

The abundance of genetic, biochemical, and structural information on PRD1 as well as the availability of several particles [e.g., (1) mature PRD1; (2) procapsid lacking genome and ATPase P9 protein; (3) icosahedral P3 shell composed of the major coat protein and minor protein P30 but lacking pentons and peripentonal capsomers; and (4) PRD1 protein-lipid membrane vesicle encapsulating a full genome] made it an ideal system for AFM analysis. Using AFM, the Abrescia group studied the deformability (stiffness, k), energy required for mechanical failure (toughness, T) and fatigue limits of the PRD1 virion and derived particles. To the best of our knowledge, this AFM study on PRD1 remains the only one for a virus in which an inner membrane vesicle surrounds densely packed dsDNA [144]. One of the most important discoveries was that full (genome-containing) or empty PRD1

particles have a similar breaking force, with the presence of DNA causing a stiffening of the mature PRD1 relative to the procapsid [144]. This implies that the mechanical stability of the PRD1 particle is independent of the presence of the genome. In contrast, the fragility (brittleness) of the P3 shell particle clearly showed a decrease in the structural reinforcement of the particle when one only considered interactions between the capsomeres in the central plate of the icosahedral facet and the P30 cementing protein. The membrane vesicle with its protein enrichment had the lowest stiffness and no discernible yield point. Thus, the membrane mostly transmits stiffening to the capsid due to DNA pressure, and its contribution to the total stiffness of mature PRD1 is limited.

To determine how the capsid and membrane vesicle collectively affect resilience, the toughness was estimated for each PRD1-derived particle and the results were compared to information available of other icosahedral dsDNA viruses. Surprisingly, only adenovirus particles can match PRD1 in toughness. Adenovirus lacks an inner membrane but has a considerable number of cementing proteins that hold the capsid together [117]. Thus, according to the study on PRD1 mechanics, the proteinaceous capsid is stiff and brittle, whereas the membrane vesicle is ductile and soft [144]. Strikingly, the high resilience of PRD1 is only reached when these two layers of materials are architecturally combined, suggesting a composite material optimized through evolution for the protection of the valuable genome.

In summary, the integration of structural and biophysical techniques has provided a unique methodological framework to unravel the PRD1 biology, which has elevated PRD1 as a model system for complex membrane-containing viruses [138].

## 8. Conclusions

After millions of years of evolution, viruses are capable of carrying out their infectious cycle with remarkable efficiency. This efficiency may result in the development of diseases with serious social and economic consequences. On the other hand, their nanometer dimensions, precise architectures, and exceptional stability properties make virus particles particularly attractive for the development of promising nanotechnological applications. In recent years, the physical properties of viruses have been revealed as the fundamental determinant of many of these functionalities. In this way, physical virology complements structure–function relationships and sheds new light on the basic principles regulating the dynamics of these remarkable molecular machines.

Currently, there is a thriving interdisciplinary research community in Spain working in this field. Since its establishment in 2010, the Spanish Interdisciplinary Network on Virus Biophysics (Biofivinet) helps connect groups within this community and with the international sphere. This network has fostered the initiation and strengthening of collaborations and knowledge interchange. The second edition of the book *Structure and Physics of Viruses* [21] is in progress. The 7th International Biofivinet Meeting will soon be held, incorporating new groups at the frontier between physics, chemistry, and biology to progressively tackle more ambitious goals in our understanding of virus assembly, disassembly, function, and engineering.

**Author Contributions:** Conceptualization, all authors; writing—original draft preparation, all authors; writing—review and editing, all authors. All authors have read and agreed to the published version of the manuscript.

**Funding:** The authors acknowledge funding by grants from the Spanish Ministry of Economy, Industry and Competitiveness (FIS2017-89549-R, MDM-2014-0377, and FIS2017-90701-REDT); the Human Frontiers Science Program (HFSPO RGP0012/2018); the Spanish State Research Agency (AEI/10.13039/501100011033); the European Regional Development Fund (PID2019-104098GB-I00, PID2019-105225GB-I00, RTI2018-096635-B-I00, PID2021-126130OB-100, PID2021-126570NB-I00, PID2021-126608OB-I00, and PID2021-126973OB-I00); the Spanish State Research Agency (CEX2021-00154-S, CEX2021-001136-S, PID2020-113287RB-I00, PID2022-136456NB-I00, RED2022-134221-T, and RED2022-134654-T); CSIC (2023AEP082 and 202120E47, LifeHub); and the Comunidad Autónoma de Madrid (P2018/NMT-4389). M.G.M. is an associate member of the Institute for Biocomputation and

Physics of Complex Systems, Zaragoza, Spain. He also acknowledges institutional support from the Ramon Areces Foundation.

**Data Availability Statement:** Not applicable.

**Acknowledgments:** The authors thank all of their past and present group members, collaborators, and colleagues who have contributed to the work reviewed here. Special thanks to all of the groups of the BioFiViNet network for supporting this interdisciplinary initiative that has boosted the field of physical virology in Spain.

**Conflicts of Interest:** The authors declare no conflict of interest. The funders had no role in the design of the study; in the collection, analyses, or interpretation of data; in the writing of the manuscript; or in the decision to publish the results.

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
