# Peer review of "Physical Virology in Spain"

_biophysica, doi:10.3390/biophysica3040041_

Round 1

Reviewer 1 Report

Comments and Suggestions for Authors

This review paper offers a view into the work of an exceptionally productive group of researchers in the field of physical virology. It describes collaborative work driven by questions that required a multi pronged approach with contributions from microbiologists, structural molecular biologists, condensed matter, and chemical physicists. The manuscript is very well written and deserves publication as is.

Author Response

We thank the reviewer for their positive appreciation of our manuscript.

Reviewer 2 Report

Comments and Suggestions for Authors

The review by Reguera et al. is a well written, comprehensive and very informative overview of the field. The authors are only kindly requested to cite the following paper on AFM study of the SARS-Cov2 virus - where this reviewer is not an author:

Kiss, B. ; Kis, Z.* ; Pályi, B. ; Kellermayer, M.S.Z. ✉ Topography, Spike Dynamics, and Nanomechanics of Individual Native SARS-CoV-2 Virions NANO LETTERS 21 : 6 pp. 2675-2680. , 6 p. (2021)  

The topic is original in the field and addresses a specific gap in the field. As a review this is a concise and detailed summary of previously published work by a number of research groups. The conclusions are consistent with the evidence and arguments presented and they address the main question posed. The references are appropriate, but one more reference was suggested.

Author Response

We thank the reviewer for their positive appreciation of our manuscript. We have added the requested reference in p. 8, line 307: “The demanding biosafety requirements for the study of these serious human respiratory pathogens require thorough adaptations of the AFM workflow [58].”

Reviewer 3 Report

Comments and Suggestions for Authors

Reguera et al. present a well written overview on physical virology in Spain and how research has progressed since the initiation of the corresponding network in 2010. The content and figures are well selected and warrant publication in Biophysica after minor revision of some of the figures.

- In some figures certain content is unreadable, e.g. in Fig. 1a+c labels in the graphs are too small. Same holds for a few other figures.

Author Response

We thank the reviewer for their positive appreciation of our manuscript. We have now reformatted Fig. 1 and panel (c) in Fig 2, so that the label visibility is improved.

Reviewer 4 Report

Comments and Suggestions for Authors

This review focuses on works from the biophysical Spanish community published in the last  ~10 years. The goal of this network of groups was to study the behaviour of viruses (mainly their capsids) by biophysical methods such as (mainly) atomic force microscopy (AFM), numeric simulations, etc. Clearly the results listed in this manuscript are directly adapted from the cited papers (including the figures). 

This review is interesting, anyway, for a larger audience, including that of "classical" structural biologists as it raise the attention on approaches that are interesting in terms of dynamics, strength of the capsids and assembly/disassembly of viruses.

Author Response

(The authors gave the same response as above.)
